# Understanding Myeloperoxidase-Induced Damage to HDL Structure and Function in the Vessel Wall: Implications for HDL-Based Therapies

**DOI:** 10.3390/antiox11030556

**Published:** 2022-03-15

**Authors:** Gunther Marsche, Julia T. Stadler, Julia Kargl, Michael Holzer

**Affiliations:** Otto Loewi Research Center, Division of Pharmacology, Medical University of Graz, Universitätsplatz 4, 8010 Graz, Austria; julia.stadler@medunigraz.at (J.T.S.); julia.kargl@medunigraz.at (J.K.); michael.holzer@medunigraz.at (M.H.)

**Keywords:** myeloperoxidase, HDL, post-translational modification, paraoxonase, cholesterol efflux capacity

## Abstract

Atherosclerosis is a disease of increased oxidative stress characterized by protein and lipid modifications in the vessel wall. One important oxidative pathway involves reactive intermediates generated by myeloperoxidase (MPO), an enzyme present mainly in neutrophils and monocytes. Tandem MS analysis identified MPO as a component of lesion derived high-density lipoprotein (HDL), showing that the two interact in the arterial wall. MPO modifies apolipoprotein A1 (apoA-I), paraoxonase 1 and certain HDL-associated phospholipids in human atheroma. HDL isolated from atherosclerotic plaques depicts extensive MPO mediated posttranslational modifications, including oxidation of tryptophan, tyrosine and methionine residues, and carbamylation of lysine residues. In addition, HDL associated plasmalogens are targeted by MPO, generating 2-chlorohexadecanal, a pro-inflammatory and endothelial barrier disrupting lipid that suppresses endothelial nitric oxide formation. Lesion derived HDL is predominantly lipid-depleted and cross-linked and exhibits a nearly 90% reduction in lecithin-cholesterol acyltransferase activity and cholesterol efflux capacity. Here we provide a current update of the pathophysiological consequences of MPO-induced changes in the structure and function of HDL and discuss possible therapeutic implications and options. Preclinical studies with a fully functional apoA-I variant with pronounced resistance to oxidative inactivation by MPO-generated oxidants are currently ongoing. Understanding the relationships between pathophysiological processes that affect the molecular composition and function of HDL and associated diseases is central to the future use of HDL in diagnostics, therapy, and ultimately disease management.

## 1. Introduction

Atherosclerosis is a chronic, lipid-driven inflammatory disease of the arteries characterized by increased oxidative stress leading to modified lipids and proteins in the vessel wall [1] accompanied by a chronic, inflammatory response that attracts cells of the innate and adaptive immune systems into the atherosclerotic plaque [2]. One of the most important enzymes released by neutrophils through degranulation is myeloperoxidase (MPO). MPO is a tetrameric, highly glycosylated, basic (PI > 10) heme protein of ~150 kDa abundant in neutrophils and monocytes [3]. The heme protein is stored in primary azurophilic granules of leukocytes and, after activation of phagocytes by various agonists, is secreted into both the extracellular milieu and the phagolysosomal compartment [4]. The release of MPO by activated leukocytes is critical to the innate immune system as it generates reactive oxidants that help kill bacteria, parasites, fungi, and other invading pathogens [5,6]. In patients with elevated levels of circulating MPO, increased cardiovascular events are observed [7]. In atherosclerotic lesions, MPO exerts its oxidative potential by using various co-substrates with H_2_O_2_ to generate reactive intermediates [8,9,10]. The presence of MPO in plaques has been associated with endothelium apoptosis, superficial erosion, and lesion rupture [11].

Of particular interest, the major HDL associated apolipoprotein A1 (apoA-I) is highly enriched (approximately 100-fold) in advanced human atherosclerotic plaques when compared with normal arterial walls [12]. ApoA-I is a selective target for myeloperoxidase-catalyzed oxidation and functional impairment in individuals with cardiovascular disease [13,14,15,16,17]. This is of particular importance, because HDL normally removes cholesterol from lipid-laden macrophages and reduces inflammation by (i) modulating immune cell function, (ii) promoting vasodilation, and (iii) enhancing endothelial barrier function [18,19,20,21]. Moreover, by modulating cholesterol content in sphingolipid- and cholesterol- enriched plasma membrane domains (lipid rafts), which play a crucial role in compartmentalizing signaling pathways, HDL suppresses immune cell activation [22,23,24]. All these favorable characteristics point to an atheroprotective role of HDL. However, apoA-I isolated from plaques is oxidatively modified by MPO, predominantly lipid-depleted, and cross-linked [12], depicting a markedly decreased cholesterol efflux capacity and lecithin-cholesterol acyltransferase activity. Therapeutic improvement of the quality and biological activity of HDL particles is therefore a challenge.

## 2. Sources and Release of MPO

MPO makes up to 5% and 1% of the total cell protein content in neutrophils and monocytes, and in some tissue macrophages in vascular lesions [1,3,25,26]. MPO is found predominantly in the primary (azurophilic) granules of neutrophils [27]. Compared to murine neutrophils, human neutrophils contain approximately 5–10 times higher amounts of MPO [28]. MPO accounts for about 1% of total cellular protein in human monocytes [29]. Macrophages can ingest extracellular MPO directly or acquire MPO by phagocytosis of apoptotic neutrophils [30]. In addition, MPO transcription in macrophages can be re-activated under some conditions [31]. Following priming and activation by inflammatory mediators, MPO can be released either by degranulation, apoptosis, necrosis [32,33] or via the extrusion of neutrophil extracellular traps [34].

## 3. MPO in Health and Disease

Under physiological and pathophysiological conditions, MPO activity is associated with neutrophil accumulation, particularly during heavy neutrophil infiltration and acute or chronic inflammation. High MPO concentrations reside in the extracellular space in close proximity to degranulated or apoptotic neutrophils. Although oxidant formation by MPO is beneficial for the immune response to invading pathogens, there is ample evidence that inappropriate stimulation of oxidant formation by MPO can lead to host tissue damage. MPO is involved in numerous physiological processes, including microbial clearance, polymorphonuclear leukocytes recruitment, neutrophil extracellular trap formation and apoptosis, as well as protein, DNA and lipid modifications [5]. The effects of MPO in the context of cancer have gained attention recently, and MPO has been associated with various pro- and antitumor properties, but most evidence indicates that MPO is a molecule that promotes tumorigenesis and progression [35].

Enzymatic reaction of MPO with H_2_O_2_ and halide or pseudohalide (thiocyanate ions) produces hypohalous acids: hypochlorous acid (HOCl), hypobromous acid, hypoiodous acid, and hypothiocyanic acid [5]. These oxidants are generally believed to be responsible for much of the antibacterial activity of neutrophils, although other oxidants including nitric oxide (NO) and peroxynitrite clearly also play important roles [6,8]. Interestingly, it was reported that MPO scavenges peroxynitrite, which may overcome the uncontrolled peroxynitrite decomposition and formation of reactive species in the inflamed vessel wall [36]. The most important reactive species produced by the MPO under physiological conditions are HOCl and hypothiocyanic acid, and the ratio of these oxidants depends critically on the concentration of thiocyanate ions [37]. The reactivity and selectivity of HOCl and hypothiocyanic acid for biological targets differ substantially, therefore thiocyanate ions have the potential to modulate both the extent and nature of oxidative damage in vivo [38]. MPO, can use thiocyanate ions and H_2_O_2_ as co-substrates to produce cyanate [17,39], promoting protein carbamylation [17,39]. Moreover, MPO was also shown to directly catalyze the oxidation of cyanide to cyanate [40], or indirectly via MPO-derived HOCl, which rapidly decomposes urea and thiocyanate, promoting cyanate formation [17]. In addition, nitrite, a product of nitric oxide radical metabolism, can be converted by MPO to a nitrogen dioxide radical, which can trigger lipid peroxidation and generate nitrotyrosine and/or nitrated lipids [38]. Alternatively, nitrite can react with HOCl to form nitrosyl chloride [41].

## 4. MPO in the Subendothelium

Elevated MPO levels have been associated with cardiovascular disease [42,43,44,45,46]. In the sub-endothelium, MPO can originate from two main sources. It either originates directly from infiltrated neutrophils or a subset of macrophages, where the granulocyte-macrophage colony-stimulating factor acts as an endogenous regulator of macrophage MPO expression [26,31]. Considering that endothelial dysfunction precedes the formation of atherosclerotic lesions, it is reasonable to assume that MPO triggers the pathophysiology of atherosclerosis early on by affecting endothelial homeostasis. Recent research has highlighted the importance of a direct interaction of MPO with the endothelium and subsequent transcytosis through the endothelium. Levels of MPO in the circulation increase highly during inflammation, mainly derived from degranulation of activated neutrophils. A unique feature of MPO is its basic nature, with an isoelectric point of >10 and a high cationic charge at physiological pH [47]. In serum, MPO is inactivated by the plasma protein ceruloplasmin, but it interacts with negatively charged components of serum and the extracellular matrix [48]. 

Studies have demonstrated that MPO released in the circulation quickly localizes in and around the endothelium [49,50]. Endothelial cells are covered by glycoproteins and proteoglycans linked to negatively charged glycosaminoglycans called glycocalyx, which forms the interface between flowing blood and the vessel wall [51]. The positively charged MPO rapidly binds to the negatively charged glycosaminoglycans on the luminal side of endothelial cells. The importance of this interaction was demonstrated when glycosaminoglycans were enzymatically removed from the glycocalyx which completely abolished binding and internalization of MPO [49,52]. The binding of MPO towards the glycocalyx lead to several detrimental effects. The cationic charge of MPO destabilizes the negatively charged endothelial glycocalyx, leading to glycocalyx collapse, allowing for neutrophil recruitment and subsequent activation [47]. Interestingly, the relatively negatively charged serum protein albumin (isoelectric point of ≈5) [53] increases the binding of MPO to endothelial cells by two-fold and the rate of transendothelial flux of MPO in cultured monolayers and intact vessels [54]. 

Albumin is the predominant plasma protein responsible for maintaining transendothelial pressure gradient and regulating the transport of fatty acids, steroids and many other proteins [55]. The albumin concentration in plasma is up to 5000 times higher than that of MPO, so the formation of this complex in the bloodstream is probably predominant [54]. Transcytosis of the albumin-MPO complex was found to be caveolae-dependent and confocal imaging indicated rapid internalization of MPO and its colocalization with albumin-labeled vesicles [49,54]. In line with the efficient transcytosis of MPO through endothelial cells, MPO-derived oxidants are enriched in endothelial and sub-endothelial compartments of vessels of patients with cardiovascular disease [25,56,57,58]. These modifications compromise cell-matrix interactions and may promote endothelial cell dysfunction.

## 5. HDL and Its Transport through the Endothelium

Recent research has provided compelling evidence for the atheroprotective activities of HDL at the vessel wall and within the extracellular space [59]. To perform these protective functions, HDL must interact with endothelial cells and gain access to the subendothelium. Endothelial cells are often considered as inert, but in fact, they are highly metabolically active far beyond the dogmatic view that metabolism drives cell growth and activity [60]. The endothelium plays an integral role in many physiological functions, including control of vasomotor tone, blood cell trafficking, hemostatic balance, permeability, proliferation, and innate and adaptive immunity [60,61]. Multiple metabolic pathways act in parallel with genetic signaling/growth factors to maintain appropriate endothelial cell behavior [60]. 

While the majority of HDL is present within the circulation, a fraction is constantly interacting with the endothelium or traveling through the extracellular space. Previous research investigated two principal pathways by which molecules can pass the endothelium barrier, being passive filtration through pores or by active transcytosis [62]. Early studies investigating the infiltration of lipoproteins into the aortic intima found that the size of the lipoprotein was the greater determinant than the interaction between the macromolecules and the arterial intimal surface. These findings suggested that the quantitatively most important mechanism for transfer of plasma lipoproteins into the arterial intima involves nonspecific molecular sieving through pores in the endothelial monolayer [63,64]. However, the so called “pore theory” had major weaknesses, including those pores large enough for lipoproteins have not been identified in endothelial monolayers and lipoproteins have not been observed in tight junctions. More recent studies have provided convincing evidence for a more active transport of lipoproteins, including HDL, through the endothelium, by a process called transcytosis [62]. 

Studies using transwell systems with human aortic endothelial cells have shown that transendothelial transport is an active process that can be blocked by lowering the temperature [65]. Moreover, when fluorescently labeled HDL was added to endothelial barriers, HDL was found only within cells but not between cells in areas associated with tight junctions [66,67]. Which pathway is used by HDL for transcytosis is not yet clear. Pharmacological inhibition of key factors involved in endocytosis of primary bovine endothelial cells and primary mouse hepatocytes suggested that HDL transcytosis is independent of clathrin and caveolin-1 [68], while it requires dynamin, which is needed for the formation of vesicles [67]. However, in human umbilical vein endothelial cells, it was reported that HDL transcytosis is clathrin-dependent, similar to LDL [66]. Further studies are needed to clarify which endocytic pathway HDL uses or whether HDL may even use different pathways depending on its composition.

At least three different proteins are involved in the current model for the transport of HDL through endothelial cells: scavenger receptor class B (SR-BI), endothelial lipase (EL), and ATP-binding cassette G1 (ABCG1) [68]. SR-BI contributes to the endocytosis, transcytosis and resecretion of HDL. Confocal and electron microscopic analyses of hepatocytes have shown that HDL particles enter cells in parallel with the movement of SR-BI [68]. Interestingly, this uptake leads to secretion of HDL-derived cholesterol at the apical side and ultimately to resecretion of HDL [69]. Furthermore, adenovirus-mediated transfection of endothelial cells with catalytically active or inactive EL has shown that EL facilitates endothelial attachment and transport by bridging and lipolysing HDL [70]. Subsequently, HDL is internalized by SR-BI and ABCG1 is transported via endosomes and multivesicular bodies before being exocytosed via an unknown mechanism [59]. 

Attached to the endothelium and during its passage through the endothelium, HDL promotes a number of vasoprotective functions. Specifically, HDL induces endothelial nitric oxide synthase (eNOS) activity resulting in vascular relaxation. HDL promotes endothelial barrier function and inhibits endothelial apoptosis. Moreover, HDL promotes endothelial repair and suppresses the expression of endothelial adhesion molecules such as vascular adhesion molecule 1 (VCAM-1) (Figure 1). In addition, HDL is an antioxidant and excellent reviews are available on this topic [21,59,71,72]. Whether HDL contributes to MPO transcytosis in a manner similar to albumin [49,54] is not known.

## 6. MPO Modifies HDL in Human Atherosclerotic Plaques

Of particular interest, apoA-I is highly enriched (approximately 100-fold) in human atherosclerotic plaques compared with normal arterial walls [12]. First evidence that MPO modifies HDL in the vessel wall was demonstrated in 2002, showing that proteins oxidized by HOCl localize with apoA-I in human atheroma [16]. Two subsequent papers then provided direct evidence that MPO oxidizes HDL in the vessel wall. These studies demonstrated that 3-chlorotyrosine content of apoA-I (a specific fingerprint of MPO oxidation) is markedly enriched in lesion-derived HDL [13,14], showing that MPO selectively targets HDL for oxidative modification in atherosclerotic lesions. 3-chlorotyrosine is an acid stable oxidation product of tyrosine that is not formed by artificial mechanisms. These characteristics make it highly useful as a specific molecular marker for MPO-catalyzed oxidation. Co-isolation of MPO with HDL-like particles extracted from human atheromas [13], and identification of a putative contact site between the apoA-I moiety of HDL and MPO [14] demonstrate that the two interact in the arterial wall. Interestingly, the binding affinity of MPO to HDL increases significantly when HDL is oxidized by the myeloperoxidase product HOCl [73], which is additional evidence that MPO promotes HDL oxidation in the human arterial wall. HOCl modification reduces the positive charge of lysine residues, through the formation of uncharged *N*-chloramines, which is expected to further increase the interaction between HDL and the positively charged MPO. 

Subsequent studies revealed that protein carbamylation is a major post-translational modification of HDL in the atherosclerotic vessel wall [17,74]. Since the content of 3-chlorotyrosine correlated significantly with the carbamyllysine content of HDL, it can be taken as evidence that the carbamylation of HDL is mediated largely by the MPO. The carbamyllysine content of lesion-derived HDL was more than 20-fold higher in comparison to 3-chlorotyrosine levels and 5 to 8-fold higher when compared to lesion LDL and increased with lesion severity. These studies clearly demonstrated that MPO catalyzes the formation of multiple oxidizing reactive species that selectively alter the structure and function of HDL in the inflamed vessel wall [3,25,26,75]. More recent studies reported that ∼20% of human apoA-I recovered from aortic plaques is modified by MPO-mediated oxidation at tryptophan residue 72 [15]. ApoA-I isolated from atherosclerotic plaques is predominantly lipid-depleted and highly oxidatively modified and cross-linked [12], suggesting low lipid binding affinity and loss of function. Studies reporting MPO mediated modification in the vessel wall are summarized in Table 1.

## 7. MPO Affects Endothelial Protective Activities of HDL

A critical factor in the development of cardiovascular disease is an alteration in endothelial cell function. This includes decreased endothelial nitric oxide availability, decreased endothelial barrier function, increased apoptosis, and adhesion molecule/chemokine expression. It is well established that HDL from healthy subjects (but also reconstituted HDL) positively affect endothelial cell functions effects [21,79]. Oxidation of HDL by MPO results in loss of HDL’s ability to activate endothelial nitric oxide synthase (eNOS) activity. HDL modified by the MPO product HOCl promotes dislocation of eNOS from the plasma membrane and perinuclear region of human endothelial cells. 

2-Chlorohexadecanal was identified as an active component mediating this inhibitory activity, which is formed during HOCl-mediated oxidative cleavage of HDL-associated plasmalogens [80]. Plasmalogens are a unique class of membrane glycerophospholipids containing a fatty alcohol with a vinyl-ether bond at the sn-1 position and represent up to 20% of the total phospholipid mass in humans [81]. Plasmalogens are a preferred target of MPO-derived reactive chlorinating species within human atheroma [82]. The generated species, alpha-chloro-fatty aldehydes and unsaturated lysophospholipids, are potent lipid mediators [82]. Unsaturated lysophospholipids are important homeostatic mediators involved in all stages of vascular inflammation [83]. Interestingly, HDL enriched with saturated lysophospholipids shows potent anti-inflammatory activities and suppresses platelet and neutrophil effector responses [84,85]. On the other side, 2-chlorohexadecanal was shown to suppress endothelial nitric oxide formation and to severely compromise endothelial barrier function, suggesting potent pro-inflammatory activities [80,82,86]. 

A subsequent study showed that MPO modification of reconstituted HDL containing only apoA-I and phosphatidylcholine also impaired eNOS activity, suggesting that oxidized apoA-I itself also affects eNOS activity [87]. MPO-modified HDL also acquires other proinflammatory functions, as seen by upregulation of VCAM-1 protein in endothelial cells and NF-κB activation and loss of anti-apoptotic properties of HDL [87]. HDL modified by the MPO product HOCl shows specific binding to lectin-like oxidized LDL receptor [16,88] and SR-BI [16]. It should be noted that one study failed to demonstrate binding of MPO-modified HDL to SR-BI [87]. MPO-modified HDL is dysfunctional in preventing an enhancement of endothelial proliferation/migration and subsequent wound healing [89,90,91] (Figure 2).

## 8. MPO Induced Oxidation of apoA-I Impairs Cholesterol Export by ATP Binding Cassette Subfamily A1 (ABCA1)

HDL transports cholesterol from peripheral cells to the liver. The ATP-binding cassette transporter A1 (ABCA1) mediates the first step of reverse cholesterol transport by transferring cellular cholesterol and phospholipids to lipid-poor HDL precursors [92]. The important role of ABCA1 in maintaining cellular lipid homeostasis is well established [93]. More recently, a novel role of ABCA1 in directly regulating inflammation has been recognized. It was shown that the interaction of apoA-I with ABCA1-expressing macrophages suppresses the ability of lipopolysaccharide to induce the inflammatory cytokines interleukin-1β, interleukin-6, and tumor necrosis factor-α [94]. The apoA-I/ABCA1 pathway functions anti-inflammatory in macrophages through activation of JAK2/STAT3, providing a direct link between the cardioprotective effects of cholesterol export from arterial macrophages and suppressed inflammation. 

In vitro and in vivo studies showed that with increasing MPO mediated oxidation of apoA-I, ABCA1-dependent cholesterol efflux from cholesterol-loaded macrophages was functionally impaired [13,14]. The presence of amphipathic helices of apolipoproteins, in addition to phospholipids, determines the high lipid binding affinity of HDL promoting efficient microsolubilization of cellular cholesterol [95]. All epitopes involved in apoA-I-mediated cellular cholesterol efflux [96] contain amino acids highly sensitive to MPO modification [97]. 

The functional consequences of oxidation or posttranslational modification of single amino acid residues are complex and not fully understood. MPO mediated carbamylation of apoA-I lysine residues does affect cholesterol efflux capacity of apoA-I [74]. In line with that observation, it was observed that lysine modification by reactive carbonyls do not affect cholesterol efflux capacity of apoA-I. Only the di-aldehydes malondialdehyde and isolevuglandins impaired the ability of apoA-I to mobilize cholesterol, the mechanism likely involves dialdehyde induced cross-linking of lysine residues, especially in repeats 9 and 10 in the C terminus [98,99]. 

Loss of ABCA1-dependent cholesterol efflux activity has been associated with increased chlorotyrosine and nitrotyrosine content of apoA-I. Specifically, chlorination of Tyr 192 and oxidation of methionine 148 have been reported to impair cholesterol efflux capacity of apoA-I, since regeneration of methionine sulfoxide using methionine sulfoxide reductase recovered the functionality of oxidized apoA-I [100]. More recent studies now suggest that MPO mediated oxidation of Trp72 residue is more relevant in impairing cholesterol acceptor activity in vivo [101]. Of note, also the treatment of apoA-I with the major MPO oxidant species hypothiocyanous acid resulted mainly in the oxidation of tryptophan residues [102]. ApoA-I containing oxTrp72-apoA-I accounts for up to 20% of apoA-I in arteries with atherosclerosis [15]. ApoA-I, which contains oxTrp72 has virtually no cholesterol acceptor activity and, consistent with lost cholesterol efflux capacity, is only very poorly lipidated [15]. Interestingly, chlorination (but not nitration) of apoA-I impairs its ability to interact directly with ABCA1 [103] in sharp contrast to the increased binding of chlorinated HDL to SR-BI as described in more detail below.

## 9. MPO Converts HDL into a High-Affinity, but Dysfunctional Ligand of the Major HDL Receptor Scavenger Receptor B1 (SR-BI)

SR-BI is an 82-kDa membrane glycoprotein with a large extracellular and two transmembrane domains [104]. SR-BI is expressed primarily in the liver and steroidogenic glands, but SR-BI is also present in other tissues and cells such as the brain, intestine, macrophages and endothelial cells. Importantly, direct evidence for the implication of SR-BI expressed on lymphatic vessels in the initial steps of reverse cholesterol transfer was provided [105]. SR-BI mediates the selective uptake of lipids from HDL into cells, a mechanism that is fundamentally different from classical receptor-mediated endocytic uptake (e.g., the LDL receptor pathway) because it involves efficient receptor-mediated transfer of lipids, but not outer coat proteins, from HDL to cells [104]. In endothelial cells, SR-BI is found on apical and basolateral plasma membranes as well as intracellularly, binding multiple macromolecules in addition to the canonical ligand HDL, including native and modified forms of LDL and HDL, as well as viruses such as SARS-CoV-2 [106]. 

In addition to mediating selective cholesteryl ester uptake from HDL [68], SR-BI also mediates free cholesterol efflux from cells [93]. In endothelial cells in particular, SR-BI exerts additional functions. In addition to transendothelial transport, binding of HDL to SR-BI triggers several intracellular signaling events. Subsequent activation of serine/threonine protein kinase B (Akt) and mitogen-activated protein kinase (MAPK) leads to downstream activation of eNOS, which results in vascular relaxation via NO [59]. Oxidation/modification of HDL consistently diminishes cholesterol efflux capacity. Only oxidative tyrosylation seems to be an exception, as tyrosyl radical modification of HDL enhances cholesterol mobilization and efflux [107]. Mutations in helices 4 and 6 of apoA-I can lead to increased high-affinity binding of reconstituted HDL particles to SR-BI [108]; however, these mutations in helices 4 and 6 of apoA-I appear to lead to “nonproductive” binding to SR-BI and cholesterol efflux is severely impaired. Methionine and tryptophan residues in apoA-I and apoA-II, respectively, are completely oxidized even at low concentrations of HOCl. Therefore, it appears that MPO modification of HDL results in high-affinity but “nonproductive” binding of HDL to SR-BI. Multiple site-specific tryptophan, methionine, and lysine modifications in apoA-I isolated from human atheroma were reported [109]. Removing the positive charge by modification/oxidation of lysine residues alters the structural characteristics of proteins, affecting protein-water interaction or protein-protein interaction. The twenty-one lysine residues of apoA-I and the N-terminus react rapidly with hypochlorous acid to form chloramines. Native plasma albumin is not recognized by SR-BI, but exposure of albumin to HOCl creates high-affinity ligands for SR-BI [110,111]. 

Interestingly, when reductively methylated albumin was oxidized with HOCl, it did not bind to SR-BI, despite oxidation of cysteine, methionine, tyrosine, histidine, and arginine residues. This supports the idea that alterations in the particular distribution of charged lysine residues of proteins increases binding affinity to SR-BI. In line with that assumption, also carbamylation of apoA-I lysine residues increases affinity to SR-BI [74]. Remarkably, one carbamyllysine residue per HDL-associated apoA-I was sufficient to increase binding affinity to SR-BI and to induce cholesterol accumulation and lipid droplet formation in macrophages [74]. Principally, it can be concluded that the binding affinity of HDL increases significantly after MPO modification. However, MPO-modified HDL does not seem to bind productively to SR-BI. Therefore, it can be assumed that MPO-modified HDL (as well as other MPO-modified lipoproteins or even MPO-modified albumin) are high affinity ligands of SR-BI and impair SR-BI mediated cholesterol uptake/efflux. 

## 10. MPO Affects Paraoxonase 1 (PON1) Activity

PON1 is an HDL-associated enzyme considered as a potential therapeutic for cardiovascular disease, but the precise mechanisms by which PON1 exerts its systemic anti-inflammatory and antioxidant effects are not well understood or defined. The HDL complex is a repository for potentially toxic, hydrophobic components of plasma, notably oxidized lipids, which may be optimal for PON1 substrates. Many lipoprotein lipids cycle through HDL during the course of metabolic processes including lipolysis of triglyceride-rich lipoproteins transport bringing HDL–PON1 into contact with exogenous, dietary lipids, which can be a rich source of oxidized lipids [112]. Moreover, the size of the HDL complex does not limit its ability to interact in close contact with tissues and cell membrane lipids. These properties of HDL would provide an attractive vehicle from which PON1 could exert a protective, antioxidant function. 

Studies in mice [113,114] and humans [115] demonstrated an antioxidant function of the protein. Interestingly, systemic PON1 activity levels can vary over orders of magnitude, whereas PON1 mass varies less [116]. In vitro studies revealed that the MPO product HOCl dose-dependently decreases PON1 activity paralleled by a dose-dependent decrease in monomeric PON1 [117]. HOCl-induced changes to PON1 activity occurred at concentrations lower than those that lead to apoA-I aggregation [117]. MPO modification of HDL leads to chlorotyrosine modification of PON1 tyrosine71, which has been detected in PON1 isolated from human atherosclerotic plaque [114]. MPO also generates the oxidant nitrogen dioxide, which drives lipid peroxidation, leading to the formation of reactive lipid dicarbonyls such as malondialdehyde and isolevuglandins which modify PON1 and lower its activity [118]. In addition, carbamylation of HDL markedly suppresses PON1 mediated paraoxonase and arylesterase activity [17]. These results demonstrate that in the human atherosclerotic plaque, where site-specific oxidation of apoA-I by chlorination, nitration and carbamylation by MPO has been demonstrated, HDL-associated PON1 is also modified by MPO (Figure 2).

## 11. MPO Affects Lecithin-Cholesteryl-Acyltransferase (LCAT) Activity

Reverse cholesterol transport is complex and involves numerous steps, including efflux of cholesterol from peripheral cells into the nascent discoidal HDL and maturation of the discoidal HDL into a cholesterol-laden spherical particle. ApoA-I is critical for both the transfer of cholesterol and phospholipids from peripheral tissues into the nascent discoidal HDL and the activation of LCAT, an enzyme critical for cholesterol esterification and maturation of HDL [119]. ApoA-I is oxidatively modified in the arterial wall at tyrosine 166 [78], a solvent-exposed region interacting with LCAT [120]. NO2-Tyr166-apoA-I was detected in atherosclerotic human coronary arteries and accounted for 8% of total apoA-I within the artery wall [78]. NO2-Tyr166-apoA-I obtained from atherosclerotic lesions exhibited a nearly 90% reduction in lecithin-cholesterol acyltransferase activity, but other concomitant oxidative modifications are expected to contribute to reduced LCAT activity [78]. In vitro data also demonstrated that carbamylation of HDL significantly impairs its ability to activate LCAT [17]. Previous studies have shown that mutation of charged residues in apoA-I results in a reduction in LCAT activity [17]. Importantly, cyanate not only targets lysine residues but also phosphatidylethanolamine [74] and interacts with cysteine groups with even higher reactivity [121]. Therefore, the decrease in the ability of HDL to activate LCAT after cyanate exposure could be the result of a change in several reactive groups in HDL-associated apoproteins, enzymes, and phospholipids.

## 12. HDL as Therapeutic Target

HDL cholesterol-raising therapeutics are currently controversial, as most phase 3 trials of therapies to chronically raise HDL cholesterol concentrations by inhibiting cholesterol ester transfer protein have shown insufficient efficacy [122,123,124,125]. It has to be noted that cholesterol ester transfer protein inhibitors are not HDL-specific drugs and in addition interfere with normal HDL metabolism. The often disappointing and conflicting results of clinical studies of HDL-cholesterol and apoA-I levels underscore the complexity and incomplete knowledge of the relationship between circulating HDL forms and the pathogenesis of coronary artery disease and therapeutic manipulation [126,127,128]. Low HDL might just be an integrated biomarker of adverse metabolic processes, such as abnormal metabolism of triglyceride-rich lipoproteins, insulin resistance, and persistent tissue inflammation. The focus of research has now shifted from increasing plasma HDL cholesterol concentrations to the function of HDL particles, which are known to be both complex and heterogeneously composed. Interventions shown to improve particular aspects of HDL functionality include lifestyle changes, especially weight loss achieved through bariatric surgery and functional diets [129,130]. Decades of research have examined immune-modulatory strategies to protect the heart after an acute myocardial infarction and prevent progression to heart failure but have failed to translate to clinical benefit. 

Of particular interest, a single intravenous dose of human apoA-I reconstituted with phosphatidylcholine (CSL111), delivered immediately postcardiac ischemia at the onset of reperfusion, improved heart function in mice [131]. Moreover, CSL111 reduced systemic and cardiac inflammatory responses by direct action on ischemic myocardium and leukocytes [132]. Studies in mice with HDL infusion [133] or increased apoA-I expression showed atheroprotective effects [134,135] motivating the initiation of infusion studies in humans with apoA-I mimetics or reconstituted HDL particles to stimulate cholesterol efflux from vascular atheroma and to reduce atherosclerotic cardiovascular disease. In summary, the preclinical data convincingly demonstrate the ability of functional HDL and lipid-poor apoA-I particles to promote the regression of atherosclerosis by effects on the number and the inflammatory state of plaque macrophages [134]. These data highlight the anti-inflammatory effects of apoA-I formulations and provide preclinical support for investigating its use in the treatment of acute coronary syndromes in the setting of primary percutaneous coronary interventions. This has prompted the concept that increasing levels of circulating apoA-I may be a new therapeutic target for preventing atherosclerotic cardiovascular disease. 

Similar to with HDL-cholesterol, there is a strong inverse observational relationship between apoA-I concentrations and risk of atherosclerotic cardiovascular disease [126]. However, the clinical translation of the pleiotropic effects of apoA-I formulations need to be validated, given that recent phase II randomized controlled trials of apoA-I infusions have not shown beneficial effects on regression of coronary atherosclerosis measured by intravascular ultrasound [136,137]. Final conclusions can be drawn following the results of an ongoing multicenter phase 3 study to evaluate the efficacy and safety of CSL112 (new formulation of apoA-I reconstituted with phosphatidylcholine) on reducing the risk of major adverse cardiovascular events in subjects with acute coronary syndrome. The current AEGIS-II study (Study to Investigate CSL112 in Patients with Acute Coronary Syndrome, NCT03473223) is addressing critical questions regarding the efficacy CSL112 infusions in patients with acute coronary syndromes to reduce recurrent severe cardiovascular events, and potential impact on hospitalization for heart failure.

## 13. Dysfunctional apoA-I/HDL in the Inflammatory Vessel Wall: Implications for Therapeutic Manipulation

As outlined above, apoA-I colocalizes with MPO in the atherosclerotic tissue [13,16] and the majority of apoA-I in arterial tissue is oxidized, carbamylated and cross-linked [13,14,15,74,78]. The function and structure (HDL-particle association) of apoA-I obtained from human arterial tissue are remarkably distinct from those in plasma (Figure 2) [12,15]. In contrast to plasma, >90% of apoA-I in normal and atherosclerotic human arterial tissue are predominantly lipid-depleted. This poor lipidation of apoA-I is explained by the fact that aorta-derived apoA-I was found to be dysfunctional, with an 80% reduction in the ability to activate lecithin-cholesterol acyltransferase and to promote cholesterol efflux. MPO oxidation of HDL also leads to the expression of endothelial adhesion molecules, reduction of eNOS synthase activity, and barrier function/wound healing of endothelial cells [80,82,87,89]. In addition, proteomic studies also showed quite conclusively that inflammation profoundly alters the composition of HDL particles in the circulation [128,138,139,140,141,142,143,144,145,146] and that plasma levels of HDL cholesterol are not necessarily functionally relevant. Therefore, studies focusing on the biological activities of plasma- or serum-derived HDL do not reflect the biology of apoA-I in the arterial wall.

These studies suggest oxidation-resistant forms of apoA-I or apoA-I mimetics as new compounds for pharmacological therapy. One study demonstrated that the apoA-I mimetic 4F promoted endothelial repair and restored the re-endothelialization impaired by the presence of MPO-modified HDL [90]. In vivo, 4F stimulated cell proliferation and re-endothelialization in the carotid artery after treatment with MPO modified-HDL in a carotid artery electric injury model but failed to do so in SR-BI deficient mice. These findings highlight that 4F promotes endothelial cell migration and demonstrates therapeutic potential against early endothelial injury in cardiovascular diseases. Intriguingly, 4F also serves as a reactive substrate for the MPO product HOCl, an antioxidant reaction that does not affect the lipid binding and cholesterol efflux capacities of the peptide [147].

Replacing 4 apoA-I tryptophan residues with phenylalanine (4WF) was shown to lead to formation of a fully functional apoA-I variant with marked resistance against oxidative inactivation by MPO-generated oxidants [109]. Importantly, the structure and function of HDL from 4WF transgenic mice was not different to HDL derived from human apoA-I transgenic mice [148].

On the other side, MPO inhibition itself could be effective to inhibit the formation of dysfunctional forms of HDL. There is accumulating evidence has implicated MPO in the pathophysiology of different diseases besides cardiovascular disease, including rheumatoid arthritis [149], kidney disease [150], pulmonary fibrosis [151], Alzheimer’s disease [152], Parkinson’s disease [153], multiple sclerosis [154], liver diseases [155] cancer [35]. Thus, MPO is an interesting drug target [156] and inhibitors have been developed but have thus far not been successful [156,157,158]. MPO is stored in a crystalline form within leukocytes granules and is released only upon activation [159], making it resistant to inactivation in its storage form. Moreover, MPO has a broad spectrum of functionality, including an important role in innate immunity and anti-microbial effects, therefore pan inhibition may cause negative effects. Therefore, instead of a global MPO inhibitor, targeted inactivation at the level of the arterial wall is ideally a more viable option. If damage has already occurred, such as in myocardial infarction and stroke, a targeted MPO inhibitor approach could be therapeutically useful to prevent further damage from the MPO-induced inflammatory responses.

## 14. Conclusions

HDL is a selective in vivo target for MPO-catalyzed oxidation in the atherosclerotic vessel wall, and extensive MPO mediated posttranslational modifications, including oxidation of tryptophan, tyrosine and methionine residues, as well as carbamylation of lysine residues are detected in apoA-I isolated from atherosclerotic plaques. Interestingly, MPO has not been detected in proteomic studies of plasma HDL [160]. This may indicate that only very small amounts of MPO circulate bound to HDL, and the sensitivity of the mass spectrometry analysis is too insensitive to detect MPO. Alternatively, it is quite possible that MPO detaches from HDL when HDL is purified by ultracentrifugation or immunoprecipitation. 

Lesion derived HDL is predominantly lipid-depleted in line with low lipid binding affinity and loss of function of lesion derived apoA-I. MPO-modified HDL is dysfunctional with respect to promote cholesterol efflux from macrophages and to activate lecithin cholesteryl-acyltransferase. MPO modified HDL lost its ability to induce endothelial nitric oxide synthesis and to promote endothelial wound healing. Therefore, it is likely that therapeutic elevation of HDL or apoA-I in the inflamed vessel wall (in the presence of high MPO activity) will not achieve the expected benefit. Unspecific antioxidant therapies have generally been ineffective, potentially due to ineffective doses but also due to interference with critical host defense and signaling processes [161]. A targeted MPO inhibitor approach could be therapeutically useful to inhibit oxidation of apoA-I in the vessel wall. MPO derived reactive oxygen species generate reactive lipid dicarbonyls, aldehydes such as malondialdehyde, 4-hydroxy-nonenal, 4-oxo-nonenal, and isolevuglandins. In the past decade small-molecule compounds have been developed to selectively and effectively scavenge these reactive lipid dicarbonyls. Preclinical data support the efficacy of novel dicarbonyl scavengers in treating or preventing disease, such as 2-aminomethylphenol [161].

Intriguingly, the apoA-I mimetic peptide 4F also serves as a reactive substrate for the MPO product HOCl, an antioxidant reaction that does not affect the lipid binding and cholesterol efflux capacities of the peptide [90]. Another approach could be to generate oxidation-resistant forms of apoA-I. It has already been shown that the replacement of four apoA-I tryptophan residues by phenylalanine leads to the formation of a fully functional apoA-I variant (4WF-apoA-I) with pronounced resistance to oxidative inactivation by MPO-generated oxidants [109,148]. Both human apoA-I and the oxidant-resistant 4WF-apoA-I delay lesion progression and promote lesion regression in LDL receptor-deficient mice, however, the 4WF isoform was not superior to the unmodified isoform in promoting lesion regression [162]. One possible reason for the lack of superiority of the 4WF isoform is that mice express much lower levels of MPO compared to humans [163]. Therefore, the 4WF isoform might have only an advantage over the wild-type isoform in a high MPO context. Further studies are needed to test the superiority of the 4WF isoform in delaying atherosclerosis progression and promoting lesion regression when compared with human wild-type apoA-I. Perhaps these new approaches will find their way into the clinic, but this cannot be foreseen at this time, as biochemical concepts are not predictive of clinical trial outcomes. The development of monoclonal antibodies that identify specific forms of dysfunctional apoA-I may be a promising approach for monitoring pathophysiological processes in the arterial wall. These monoclonal antibodies could be used to evaluate targeted HDL/apoA-I therapies aimed at attenuating or even preventing MPO-induced HDL modifications.

## Figures and Tables

**Figure 1 antioxidants-11-00556-f001:**
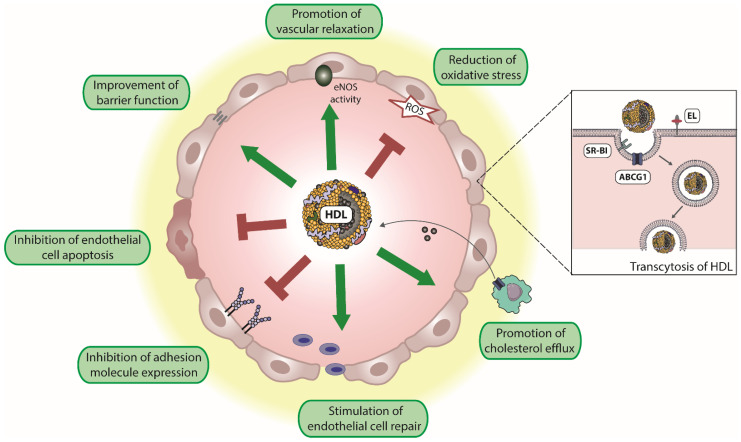
Endothelium-protective activities of HDL. HDL particles exert several protective effects on the endothelium, including reduction of reactive oxygen species (ROS), the improvement of endothelial barrier function, and promotion of vascular relaxation by increasing endothelial nitric oxide synthase (eNOS) activity. Moreover, HDL inhibits endothelial cell apoptosis, suppresses the expression of endothelial adhesion molecules, and stimulates endothelial cell repair. In addition, HDL promotes reverse cholesterol transport, by uptake of cholesterol from macrophages and other peripheral cells. Transendothelial transport of HDL is mediated by scavenger receptor B1 (SR-BI), ATP-binding cassette G1 (ABCG1), and endothelial lipase (EL).

**Figure 2 antioxidants-11-00556-f002:**
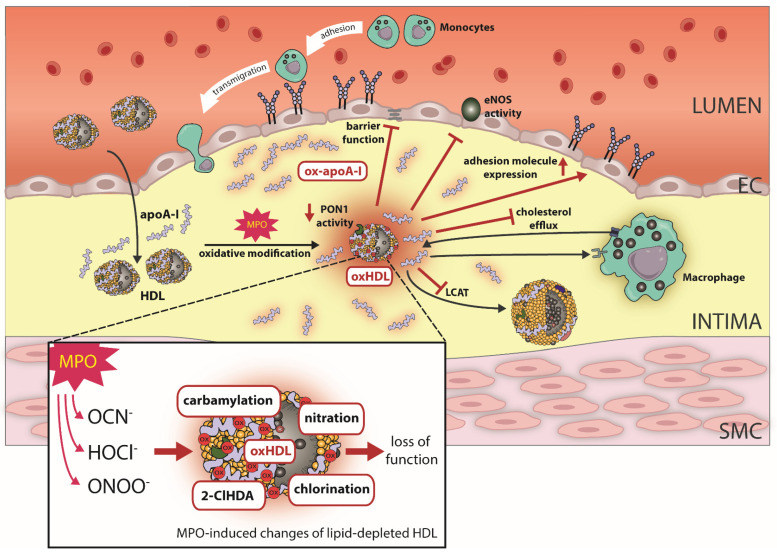
Effect of MPO-induced oxidative modifications on HDL function. In the atherosclerotic vessel wall, HDL/apoA-I is a target for MPO-catalyzed oxidation. Specifically, the MPO products hypochlorus acid (HOCl^−^), cyanate (OCN^−^) and peroxynitrite (ONOO^−^) lead to chlorination, carbamylation, nitration and the formation of the plasmalogen oxidation product 2-Chlorohexadecanal (2-ClHDA). Oxidative modifications of HDL by MPO results in loss of HDL’s ability to activate endothelial nitric oxide synthase (eNOS). Moreover, MPO modified HDL compromises endothelial barrier function and upregulates endothelial adhesion molecule expression. Further, MPO-catalyzed oxidation of HDL impairs cholesterol efflux capacity via ABCA1, whereas affinity for SR-BI increases. MPO also targets PON1 and leads to decreased activity. Oxidative modifications of apoA-I result in a profoundly decreased activity of LCAT. ApoA-I, apolipoprotein A1; MPO, myeloperoxidase; ABCA1, ATP-binding cassette transporter A1.

**Table 1 antioxidants-11-00556-t001:** Summary of studies evaluating MPO-mediated protein oxidation in humans.

Study	Subjects	Sample Type	3-Chlorotyrosine/Tyrosine (µmol/mol)	*p*
Zheng et al. [14]	Healthy (*n* = 44)	plasma protein	1.6 (0.6–2.4)	
	Coronary vascular disease (*n* = 45)	plasma protein	1.9 (1.3–3.1)	0.070
	Healthy (*n* = 44)	plasma apoA-I	186 (114–339)	
	Coronary vascular disease (*n* = 45)	plasma apoA-I	500 (335–650)	<0.001
	Aorta (*n* = 10)	total protein	63 (25–128)	
	Aorta (*n* = 10)	apoA-I	678 (299–1311)	<0.001
	Aortic atherosclerotic tissue (*n* = 22)	total protein	232 (111–431)	
	Aortic atherosclerotic tissue (*n* = 22)	apoA-I	3930 (1679–7005)	<0.001
Bergt et al. [13]	Healthy (*n* = 8)	plasma HDL	3 ± 2	
	Coronary artery disease (*n* = 9)	plasma HDL	39 ± 7	<0.0001
	Atherosclerotic patients (*n* = 17)	plasma HDL	22 ± 7	
	Atherosclerotic patients (*n* = 10)	lesion HDL	177 ± 27	0.0001
**Study**	**Subjects**	**Sample Type**	**Homocitrulline/lysine (µmol/mol)**	** *p* **
Wang et al. [39]	Healthy (*n* = 300)	plasma protein	0.11 (0.00–0.52)	
	Coronary artery disease (*n* = 137)	plasma protein	0.3 (0.02–0.80)	<0.001
	Peripheral artery disease (*n* = 62)	plasma protein	0.33 (0.0–0.95)	<0.001
	Coronary and Peripheral artery disease (*n* = 49)	plasma protein	0.39 (0.15–0.96)	<0.001
	Control, event-free (*n* = 275)	plasma protein	0.18 (0.00–0.84)	
	Revascularization (*n* = 224)	plasma protein	0.33 (0.02–0.93)	<0.001
	Myocardial infarction or stroke (*n* = 38)	plasma protein	0.33 (0.04–0.84)	<0.001
	Death (*n* = 55)	plasma protein	0.34 (0.03–0.94)	<0.001
Holzer et al. [74]	Atherosclerotic patients (*n* = 5)	plasma HDL	215 ± 27	
	Healthy control vessel (*n* = 5)	lesion HDL	229 ± 32	
	Atherosclerotic lesion, Type I (*n* = 5)	lesion HDL	1098 ± 189	0.001
	Atherosclerotic lesion, Type II/III (*n* = 5)	lesion HDL	1238 ± 84	0.0002
	Atherosclerotic lesion, Type IV/V (*n* = 5)	lesion HDL	2270 ± 283	<0.0001
**Study**	**Subjects**	**Sample Type**	**3-NO_2_-Tyrosine (µmol/mol)**	** *p* **
Pennathur et al. [76]	Healthy (*n* = 5)	total protein	8 ± 6	
	Healthy (*n* = 5)	plasma HDL	57 ± 10	0.010
	Healthy (*n* = 8)	plasma HDL	68 ± 7	
	Coronary vascular disease (*n* = 9)	plasma HDL	136 ± 11	<0.01
	Atherosclerotic patients (*n* = 17)	plasma HDL	104 ± 39	
	Atherosclerotic patients (*n* = 10)	lesion HDL	619 ± 178	<0.01
Zheng et al. [16]	Healthy (*n* = 44)	plasma protein	6 (4–8)	
	Coronary vascular disease (*n* = 45)	plasma protein	9 (6–13)	<0.001
	Healthy (*n* = 44)	plasma apoA-I	438 (335–598)	
	Coronary vascular disease (*n* = 45)	plasma apoA-I	629 (431–876)	0.005
	Human aorta (*n* = 10)	total protein	55 (24–143)	
	Human aorta (*n* = 10)	lesion apoA-I	401 (185–637)	<0.001
	Aortic atherosclerotic tissue (*n* = 22)	total protein	108 (51–346)	
	Aortic atherosclerotic tissue (*n* = 22)	lesion apoA-I	2340 (1665–5050)	<0.001
Shao et al. [77]	Atherosclerotic patients (*n* = 11)	plasma HDL	19 ± 13	
	Atherosclerotic patients (*n* = 8)	lesion HDL	242 ± 160	0.006
**Study**	**Subjects**	**Sample Type**	**3-NO_2_-Tyrosine^166^ (% of Total)**	** *p* **
DiDonato et al. [78]	Healthy (*n* = 5)	plasma protein	0.14 ± 0.02	
	Healthy (*n* = 5)	plasma HDL	n.d.	
	Healthy (*n* = 5)	plasma LPD	0.13 ± 0.02	
	Healthy (*n* = 5)	plasma apoA-I	0.12 ± 0.18	n.r.
	Atherosclerotic patients (*n* = 5)	tissue protein	7.79 ± 5.57	
	Atherosclerotic patients (*n* = 5)	lesion HDL	0.23 ± 0.37	
	Atherosclerotic patients (*n* = 5)	lesion LPD	9.05 ± 4.86	n.r.

LPD, lipoprotein-deficient fraction; n.r., not reported.

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
