# Peer review of "Understanding Myeloperoxidase-Induced Damage to HDL Structure and Function in the Vessel Wall: Implications for HDL-Based Therapies"

_antioxidants, 2022, doi:10.3390/antiox11030556_

Round 1

Reviewer 1 Report

After reviewing the manuscript of "Understanding myeloperoxidase-induced damage to HDL structure and function in the vessel wall: implications for HDL-based therapies". The overall quality of the manuscript is well. I have these following minor comments.

Result:

-Some of the sections in the main text were merely a list of the references and lack of summarization from the authors.

Discussion:

- Consider adding a concluding section to integrate the current evidence and also to present the interpretation of these results from the authors' perspective. 

Language/formatting:

- Please consider breakdown the sections into different paragraphs to avoid a cluster of references for a better understanding. 

-The table style is confusing. Consider revision.

- Minor grammatical and syntax errors (e.g. was/were, repeated structure, etc.).

Author Response

After reviewing the manuscript of "Understanding myeloperoxidase-induced damage to HDL structure and function in the vessel wall: implications for HDL-based therapies". The overall quality of the manuscript is well. I have these following minor comments.

We are glad to hear that our manuscript was positively received by the reviewer and we are happy to consider the helpful comments.

Result:

-Some of the sections in the main text were merely a list of the references and lack of summarization from the authors.

We have now tried to better summarize the literature in the corrected version of the manuscript.

Discussion:

- Consider adding a concluding section to integrate the current evidence and also to present the interpretation of these results from the authors' perspective. 

We have a detailed Conclusion in the revised manuscript, where we also give our interpretation of the data (lines 1063-1176).

Language/formatting:

- Please consider breakdown the sections into different paragraphs to avoid a cluster of references for a better understanding. 

According to the reviewer’s suggestion, we have included breakdowns into the different paragraphs

-The table style is confusing. Consider revision.

We thank the reviewer for pointing out this issue; we have revised the table according to the reviewer’s suggestion.

- Minor grammatical and syntax errors (e.g. was/were, repeated structure, etc.).

We sincerely apologize to the reviewers for these mistakes; we have tried to correct the manuscript accordingly.

Reviewer 2 Report

In this review, the authors provide an overview of  the pathophysiological consequences of myyleoperoxidase-induced changes in the structure and function of HDL and discuss possible therapeutic implications. The review is very well structured and the Figures are of high quality. There is a logical flow of data.

Minor comments:1. Section 12. The authors state: 'HDL therapeutics are currently controversial, as several phase 3 trials of therapies to chronically increase HDL cholesterol concentrations by cholesterol ester transfer protein inhibition have failed or demonstrated insufficient efficacy’. First all, CETP inhibitors are not HDL-specific drugs and in addition interfere with normal HDL metabolism. Secondly, reference 125 can not be cited as a negative study. There was a significant effect on the primary endpoint. The reasons why this drug did not make it to the market are very specific but in any case, REVEAL is not a negative trial. 2. There is only one study that is truly investigating the original HDL hypoothesis and that is the AEGIS-II trial, also mentioned in the paper. The outcome of this trial is unknown at present. Biochemical concepts are not predictive of the outcome of clinical trials with hard clinical endpoints. This is also applicable to the strategies proposed in this review. 3. Low HDL may be an integrated biomarker of adverse metabolic processes including abnormal metabolism of triglyceride rich lipoproteins, insulin resistance, and ongoing tissue inflammation. The HDL hypothesis may simply be wrong. This does not exclude that certain pharmacological approaches related to HDL particles will have an effect.

Author Response

In this review, the authors provide an overview of  the pathophysiological consequences of myyleoperoxidase-induced changes in the structure and function of HDL and discuss possible therapeutic implications. The review is very well structured and the Figures are of high quality. There is a logical flow of data.

We are glad to hear that our manuscript was positively received by the reviewer and we are happy to consider the helpful comments.

Minor comments:1. Section 12. The authors state: 'HDL therapeutics are currently controversial, as several phase 3 trials of therapies to chronically increase HDL cholesterol concentrations by cholesterol ester transfer protein inhibition have failed or demonstrated insufficient efficacy’. First all, CETP inhibitors are not HDL-specific drugs and in addition interfere with normal HDL metabolism. Secondly, reference 125 can not be cited as a negative study. There was a significant effect on the primary endpoint. The reasons why this drug did not make it to the market are very specific but in any case, REVEAL is not a negative trial. 2. There is only one study that is truly investigating the original HDL hypoothesis and that is the AEGIS-II trial, also mentioned in the paper. The outcome of this trial is unknown at present. Biochemical concepts are not predictive of the outcome of clinical trials with hard clinical endpoints. This is also applicable to the strategies proposed in this review. 3. Low HDL may be an integrated biomarker of adverse metabolic processes including abnormal metabolism of triglyceride rich lipoproteins, insulin resistance, and ongoing tissue inflammation. The HDL hypothesis may simply be wrong. This does not exclude that certain pharmacological approaches related to HDL particles will have an effect.

We thank the reviewer for that very important comment. We agree, we have rewritten that sentence into “HDL cholesterol-raising therapeutics are currently controversial, as most phase 3 trials of therapies to chronically raise HDL cholesterol concentrations by inhibiting cholesterol ester transfer protein have shown insufficient efficacy [122–125]. It has to be noted that cholesterol ester transfer protein inhibitors are not HDL-specific drugs and in addition interfere with normal HDL metabolism”.

We now also included the following sentence into the Conclusion (line 1171) “Perhaps these new approaches will find their way into the clinic, but this cannot be foreseen at this time, as biochemical concepts are not predictive of clinical trial outcomes.”